# Binding mechanism of oseltamivir and influenza neuraminidase suggests perspectives for the design of new anti-influenza drugs

**Jiaye Tao**[1☯], **Heping Wang**[2☯], **Wenjian Wang**[2☯], **Na Mi**[1], **Wei Zhang**[1], **Qiujia Wen**[1], **Jiajing Ouyang**[1], **Xinyun Liang**[1], **Min Chen**[1], **Wentao Guo**[1], **Guoming Li**[1], **Jun Liu**[1], **Hanning Zhao**[1], **Xin Wang**[1], **Xuemeng Li**[1], **Shengjun Feng**[1], **Xinguang Liu**[3]*, **Zhiwei He**[4]*, **Zuguo Zhao**[1]*

**1** Department of Microbiology and Immunology of Basical Medicine of Guangdong Medical University, Dongguan, Guangdong Province, China, **2** Department of Respiratory diseases of Shenzhen Children's Hospital, Shenzhen, Guangdong Province, China, **3** Guangdong Provincial Key Laboratory of Medical Molecular Diagnostics, Institute of Aging Research, Guangdong Medical University, Dongguan, Guangdong Province, China, **4** Department of Sino US joint Cancer Institute of Guangdong Medical University, Dongguan, Guangdong Province, China

☯ These authors contributed equally to this work.
* xgliu@gdmu.edu.cn (XL); hezhiwei@gdmu.edu.cn (ZH); zhaozuguo@gdmu.edu.cn (ZZ)

**Data Availability Statement:** All data are included in the article and/or S1 Appendix.

**Funding:** This work was supported by grants from Discipline Construction Project of Guangdong

## Abstract

Oseltamivir is a widely used influenza virus neuraminidase (NA) inhibitor that prevents the release of new virus particles from host cells. However, oseltamivir-resistant strains have emerged, but effective drugs against them have not yet been developed. Elucidating the binding mechanisms between NA and oseltamivir may provide valuable information for the design of new drugs against NA mutants resistant to oseltamivir. Here, we conducted large-scale (353.4 μs) free-binding molecular dynamics simulations, together with a Markov State Model and an importance-sampling algorithm, to reveal the binding process of oseltamivir and NA. Ten metastable states and five major binding pathways were identified that validated and complemented previously discovered binding pathways, including the hypothesis that oseltamivir can be transferred from the secondary sialic acid binding site to the catalytic site. The discovery of multiple new metastable states, especially the stable bound state containing a water-mediated hydrogen bond between Arg118 and oseltamivir, may provide new insights into the improvement of NA inhibitors. We anticipated the findings presented here will facilitate the development of drugs capable of combating NA mutations.

## Author summary

Influenza virus neuraminidase (NA), a viral membrane glycoprotein, plays an important role in the interactions with host cell surface receptors. The emergence and spread of influenza mutants resistant to neuraminidase inhibitors (NAIs), such as oseltamivir, has been of great concern. Despite many improvements to NAIs, no new first-line NAIs are

Medical University (4SG21279P to XL),
Disciplinary Construction of Posts for Zhujiang
Scholars (4SG21005G to ZH), Shenzhen Fund for
Guangdong Provincial High-level Clinical Key
Specialties (SZGSP012 to WW) and Shenzhen Key
Medical Discipline construction Fund (SZXK032 to
WW) and General project of Guangdong Natural
Science Foundation (2021A1515011403 to XL).
The funders had no role in study design, data
collection and analysis, decision to publish, or
preparation of the manuscript.

**Competing interests:** The authors have declared
that no competing interests exist.

currently in clinical use. Although there have been previous molecular dynamics simulation studies on the binding and dissociation process of oseltamivir-NA, we discovered new binding pathways and states of oseltamivir through larger-scale simulations and more systematic analysis, which may provide new ideas for the improvement of oseltamivir and even a series of NAIs. In our study, we strongly demonstrate that a detailed understanding of the drug–receptor association process is of fundamental importance for drug design and provide methodological references for the improvement of other drugs.

## Introduction

Influenza A virus has long been a great threat to human health, and there have been many influenza outbreaks leading to considerable numbers of deaths throughout history [1–3]. The neuraminidase (NA) of influenza virus binds sialic acid on the host cell and disrupts the terminus of N-acetyl neuraminic acid (Neu5Ac) α-2,3- and α-2,6- linked sialic acid moieties to facilitate the release of new virus particles from the surface of infected host cells [4,5]. Influenza viruses can be divided into two groups based on phylogenetic analysis of the neuraminidase gene: group-1 (N1, N4, N5, N8) and group-2 (N2, N3, N6, N7, N9) [6]. NA inhibitors, designed based on group-2 NA structures, can inhibit influenza A and B virus NA, thus inhibiting the release of viruses from host cells [7–11]. At present, oseltamivir (Tamiflu), zanamivir (Relenza) and peramivir (Rapivab) are FDA-approved NA inhibitors (NAIs) for the treatment of influenza, of which oseltamivir is the most widely used. However, with the wide use of oseltamivir in the clinic, the emergence and worldwide spread of oseltamivir-resistant mutants have attracted close attention [7,12]. Therefore, there is an urgent need to design new antiviral drugs. Elucidating the binding mechanism between influenza NA and its inhibitor and identifying the key metastable state during their binding may help to improve the structure of inhibitors and deal with resistant mutations of NA.

Although many crystal structures of oseltamivir-NA holo complexes have been resolved, these crystal structures provide only frozen snapshots of the NA-NAI interactions at the catalytic (1˚) site. The 150 loop (residues 147–152) and 430 loop (residues 429–433) around the NA 1˚ site exhibit high flexibility [13–16]. The 150 loop can shift between open and closed states, and the open 150 loop can form the so-called 150 cavity [13,6]. At present, drugs for the 150 cavity that can effectively fight oseltamivir-resistant viruses have been designed [17]. In the vicinity of the 1˚ site, another sialic acid binding site, a secondary (2˚) binding site, exists. The 2˚ site is a shallow pocket formed by three loop structures on the NA surface; it mediates hemagglutination activity but cannot be inhibited by NAIs [4,18–21]. By analyzing the distribution of chloride ions along the NA surface, Durrant *et al.* speculated that oseltamivir may bind to the 2˚ site first and then be transported to the 1˚ site by electrostatic attraction [13]. However, this speculation had not been proven.

Some studies have tried to reveal the binding and dissociation of oseltamivir NA using molecular dynamics (MD) [22–25], but the complete binding pathway, the interaction between oseltamivir and NA during binding, and the role of the 2˚ site in the binding of oseltamivir to the 1˚ site remain unknown. In this study, we performed a large-scale free-binding MD simulation to study the complete binding process between oseltamivir and N9-type NA and applied meta-eABF [26–28] with a Markov State Model (MSM) [29–31] to analyze the metastable state, free-energy landscape and binding pathways. Considering the computational cost, in the large-scale MD simulation, we used the NA monomer for the simulation and limited the diffusion range of oseltamivir to the 1˚ and 2˚ sites and their nearby areas by the

position restraint method. At the same time, we used the NA tetramer for a small-scale simulation to verify the results of the NA monomer simulation. Our study showed that oseltamivir entered the 1˚ site mainly through five binding pathways and the previously proposed hypothesis that oseltamivir can be transferred from the 2˚ site to the 1˚ site was verified in this study. Some new metastable states during the oseltamivir-NA binding process were observed in this study, which may help improve oseltamivir or the design of new anti-influenza drugs.

## Results and discussion

### Overview of the simulations

Oseltamivir was designed based on the 1˚ site in the crystal structure of group-2 NA [9]. Therefore, the N9 NA in the crystal structure (PDB ID: 2QWK) [32] was used as the receptor to study the oseltamivir-NA binding process. To reduce the computational overhead only the NA monomer was used to perform large-scale MD simulations, and the NA tetramer was used to perform small-scale simulations. Detailed information on the MD systems can be found in S1 Appendix, Section 1. Although oseltamivir carboxylate is the active metabolite of oseltamivir and interacts with NA, for convenience, oseltamivir carboxylate was replaced with oseltamivir in the following text. In both NA monomer and tetramer systems, the motion range of oseltamivir was limited to a hemispherical range with a radius of 27 Å above the 1˚ site (Fig 1A), which included the 1˚ and 2˚ sites and their adjacent regions.

We performed a total of 589 free-binding simulations (600 ns each) for oseltamivir and NA monomers, for a total simulation time of 353.4 μs. When the RMSD of the oseltamivir molecule is less than 2 Å in a trajectory with respect to that of the oseltamivir molecule in the crystal structure (PDB ID: 2QWK), this trajectory is considered positive (Fig 2). There were 112 positive trajectories (the total time was 67.2 μs), resulting in a positive rate of 19%. The RMSD of oseltamivir in the trajectories showed that oseltamivir can reach the final stable binding state as early as 10 ns or as late as 585 ns (Fig 2). In our simulations, once oseltamivir binds to the 1˚ site and achieves the final stable binding state, it remains steady and does not leave the 1˚ site.

To analyze the detailed binding process and metastable states of oseltamivir and NA, we employed PyEMMA [33] to construct the MSM for all 112 positive trajectories. Our model was verified to be Markovian by the implied time scales and the Chapman-Kolmogorov test. At the same time, the free-energy landscape of binding process was obtained by a meta-eABF simulation (S1 Appendix, Section 2) implemented in NAMD [34], which revealed the free-energy profile of oseltamivir when moving along the surface of NA.

### Identification of the metastable states

Ten metastable states (S0-S9) were identified based on the analysis of the free-energy landscape (please see below), the results of the implied timescales and the Chapman-Kolmogorov test. The detailed interactions between oseltamivir and NA in each metastable state can be found in Figs 3 and S1.

**Metastable state S1:** This state was also called the unbound state, in which oseltamivir was surrounded by solvent molecules and was more than 5 Å from the surface of NA.

**Metastable state S0:** In this state, oseltamivir adopted some interesting configurations (Fig 3). To find the proportion of each typical conformation in S0, we used the gromos algorithm [35] in GROMACS [36] to cluster them. Our results showed that oseltamivir in S0 was mainly located in two regions. We named the conformations in these two regions S0A, S0B, which accounted for 33.5% and 14.5% of S0. In S0A, oseltamivir was located in the 2˚ site. Since there is no report on the crystal structure of oseltamivir at the 2˚ site of NA, we compared the S0A with the cocrystal structure of sialic acid-NA (PDB ID: 1MWE) [37] (Fig 4). Due to the steric

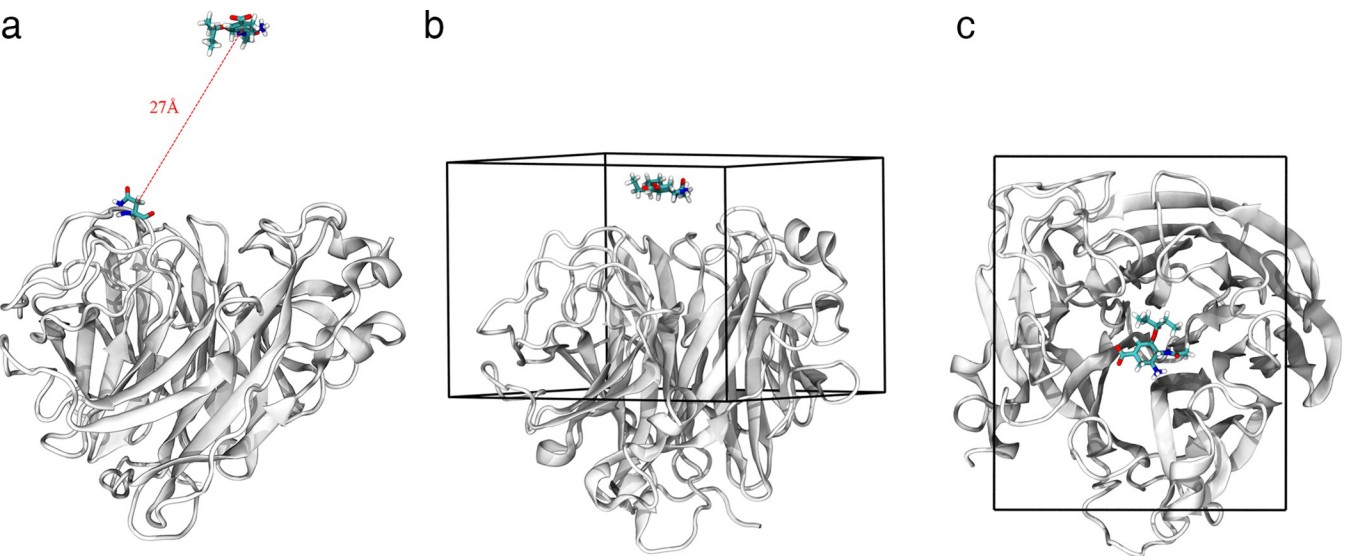

**Fig 1. Sampling space for oseltamivir in the simulations.** (a) The diffusion region of oseltamivir in the free-binding MD simulation; oseltamivir was restricted in a space centered on the beta carbon of N346 with a radius of 27 Å. (b) and (c) Side and top views of the sampling region (cuboid box with a size of 46 Å×38 Å×32 Å) for the meta-eABF simulation.

hindrance of the protonated amino group of oseltamivir and the weak flexibility of the carboxylate of oseltamivir compared with that of sialic acid, the conformation of oseltamivir was quite different from that of sialic acid at the 2˚ site though with the same orientation as sialic acid. The distal hydroxyl oxygen (O9) of the 6-triol group of sialic acid interacted with the ε-amino group of K432 and the other hydroxyl oxygen (O8) of sialic acid interacted with the

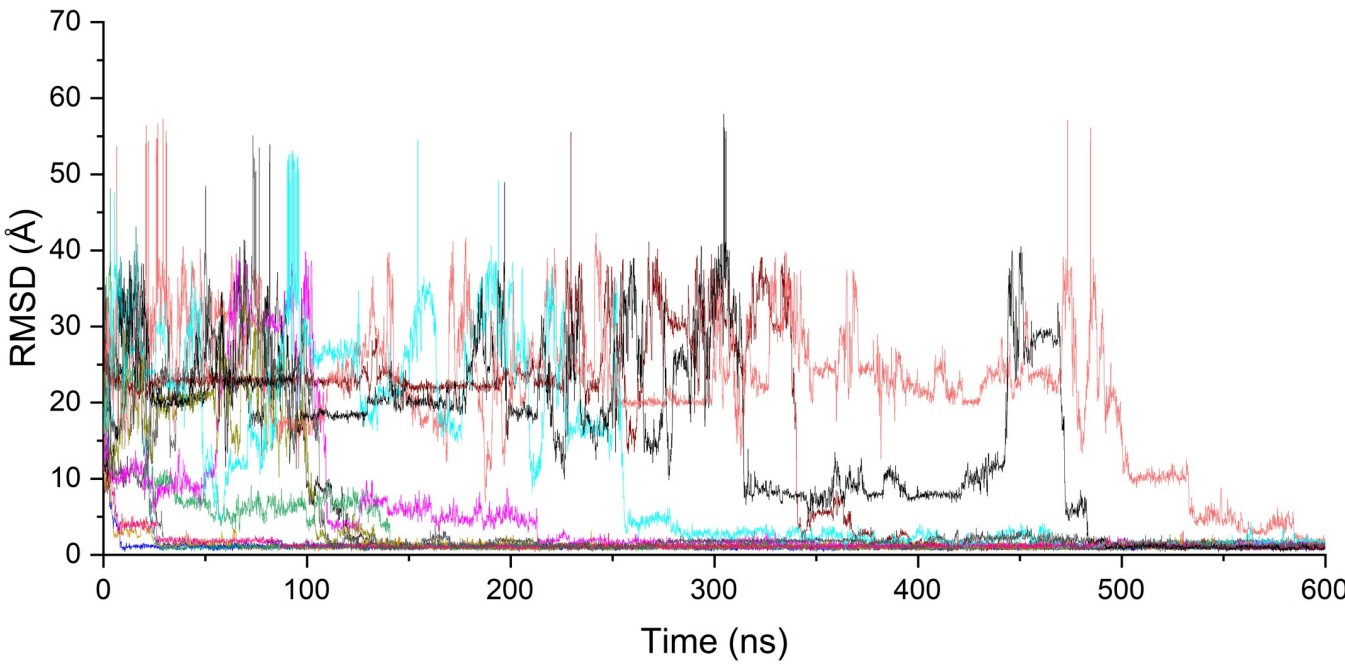

**Fig 2. Typical RMSDs of oseltamivir in the trajectories produced binding events.** The RMSD of heavy atoms of oseltamivir was calculated by referring those of oseltamivir of the crystal structure.

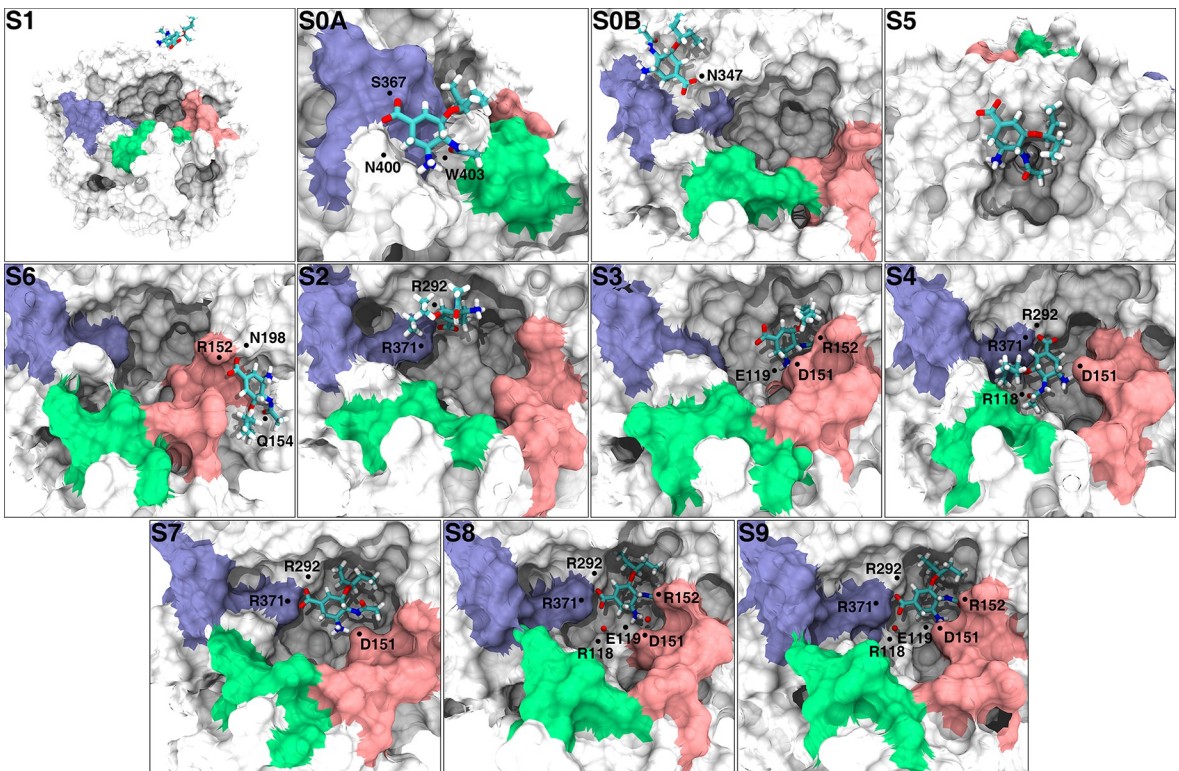

**Fig 3. Interaction of oseltamivir with NA in metastable S0-S9.** The position of the residue that participated in H-bond interactions with oseltamivir is marked with black dots. The red spheres in S8 and S9 represent the oxygen atoms of water molecules (the hydrogen atoms of water molecules are not shown), which participated in the formation of water-mediated H-bonds. The 150 loop, 370 loop and 430 loop regions are highlighted in pink, ice blue and green, respectively.

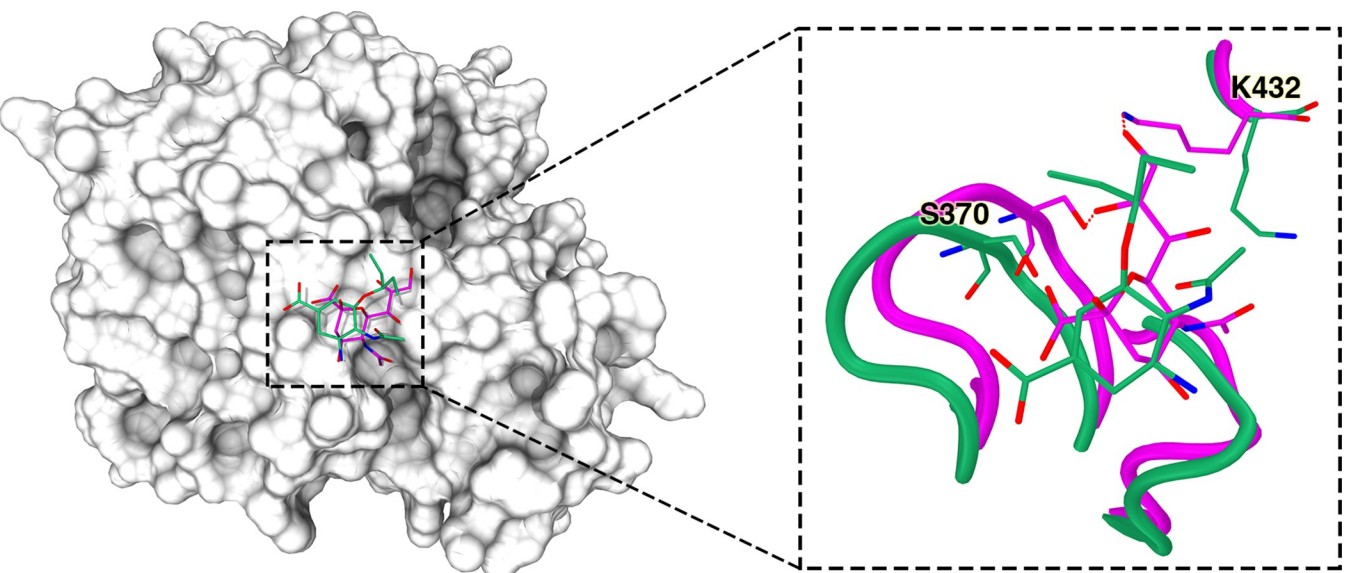

**Fig 4. Comparison of the conformations of sialic acid and oseltamivir at the secondary binding site.** The position outlined by the black dotted square in the left panel is the secondary binding site of NA. Emerald green in the right panel represents oseltamivir and NA in the simulation, and magenta represents sialic acid and NA in the crystal structure (PDB ID: 1MWE). Hydrogen bonds between sialic acid and partial residues in the secondary binding site are shown as dashed red lines.

hydroxyl oxygen of S370. However, the pentyl group at the corresponding position (C6) of oseltamivir did not form a H-bond with K432 or S370, which may be related to the conformational instability of oseltamivir at this site. In the typical conformation of S0A, oseltamivir formed H-bonds with S367, N400 and W403 and hydrophobic interactions with S367, S370 and N400.

In S0B, oseltamivir was located in the electrostatic funnel proposed by Le *et al* [22]. P326, R327, G343, N344, N346, N347, G348, and A369 were the main residues that interacted with oseltamivir. Oseltamivir adopted a variety of conformations at this site. In the typical conformation of S0B, oseltamivir interacted with only N347.

**Metastable state S5:** In this state, oseltamivir was located in the shallow pocket composed of the 250 loop, the 270 loop and partial β sheets adjacent to the 250 loop (Fig 3), and formed hydrophobic interactions with P249, A250, K273 and I275. In addition, we noticed that oseltamivir was located at one of the antigen epitopes (consists of A247, T248, P249 and A272) of N1-type NA recognized by neutralizing antibodies but not the antigen epitopes (consists of A247, T248, P249 and A272) in N9-type NA [38–40].

**Metastable state S6:** In this state (Fig 3), oseltamivir mainly interacted with the 150 loop. The carboxylate moiety of oseltamivir faced the center of the 1˚ site, and an oxygen atom on the carboxylate group formed two H-bonds with the guanidine group of R152 on the 150 loop and one H-bond with the amide of the N198 side chain. The acetyl group of oseltamivir formed one H-bond with the N of the peptide bond of Q154. The pentyl group of oseltamivir had hydrophobic interactions with the side chain of S153.

**Metastable state S2:** In this state, oseltamivir entered the 1˚ site with an inclined posture. The carboxylate group of oseltamivir formed H-bonds with the side chains of R292 and R371, which was similar to that observed in previous studies [22,23].

**Metastable state S3:** Compared with the conformation of oseltamivir in the crystal structure, the carboxylate and pentyl groups of oseltamivir in S3 were lifted up and pointed toward the solvent due to the absence of H-bonds between the guanidine groups of the R292 and R371 side chains that can stabilize the conformation of oseltamivir (Fig 3). The acetyl group and protonated amino group of oseltamivir formed H-bonds with the guanidine group of R152 and the carboxylate group of D151 and E119, respectively. The pentyl group and acetyl group of oseltamivir had hydrophobic interactions with the side chains of R224 and W178, respectively.

**Metastable state S4:** The cyclohexane plane of oseltamivir in this state was nearly parallel to that in the crystal structure, but the conformation of the entire oseltamivir was almost perpendicular to that in the crystal structure (Fig 3). In S4, the acetyl group of oseltamivir formed a H-bond with the guanidine group of R118, and the carboxylate group of oseltamivir formed a H-bond with the guanidine group of R371 and a H-bond with the guanidine group of R292 simultaneously. The pentyl group of oseltamivir had a hydrophobic interaction with the side chains of W403 and R371, and the acetyl group of oseltamivir had a hydrophobic interaction with the side chain of R430. By manually analyzing the trajectories, we found that the duration of S4 is usually 10 ns to 100 ns.

**Metastable state S7:** In this state, oseltamivir did not enter the 1˚ site completely. R152 faced the outside of the 1˚ site without covering the acetyl group of oseltamivir (Fig 3). The carboxylate group of oseltamivir formed a bidentate H-bond with the guanidine group of R371 and a H-bond with the guanidine group of R292. The protonated amino group of oseltamivir formed a H-bond with the carboxylate group of D151. The pentyl group of oseltamivir had hydrophobic interactions with the side chains of A246, N294 and R292.

**Metastable state S8 and S9:** In S8 (Fig 3), the configuration of oseltamivir was very close to the final stable binding state (S9) except the water-mediated H-bond between the carboxylate

group of oseltamivir and the side chain of D151. In S9 (Fig 3), the RMSD of oseltamivir was less than 2 Å from the oseltamivir in the crystal structure. The acetyl group of oseltamivir formed a H-bond with the guanidine group of R152. The carboxylate group of oseltamivir formed a bidentate H-bond with the guanidine group of R371 and a H-bond with the guanidine group of R292, and the protonated amino group of oseltamivir formed H-bonds with the side chains of D151 and E119. The pentyl group of oseltamivir had hydrophobic interactions with W178, I222, R224, A246, and E276; and the acetyl group of oseltamivir had hydrophobic interactions with W178 and I222. Notably, in the crystal structure, there was a direct H-bond between the carboxylate group of oseltamivir and the side chain of R118 in the crystal

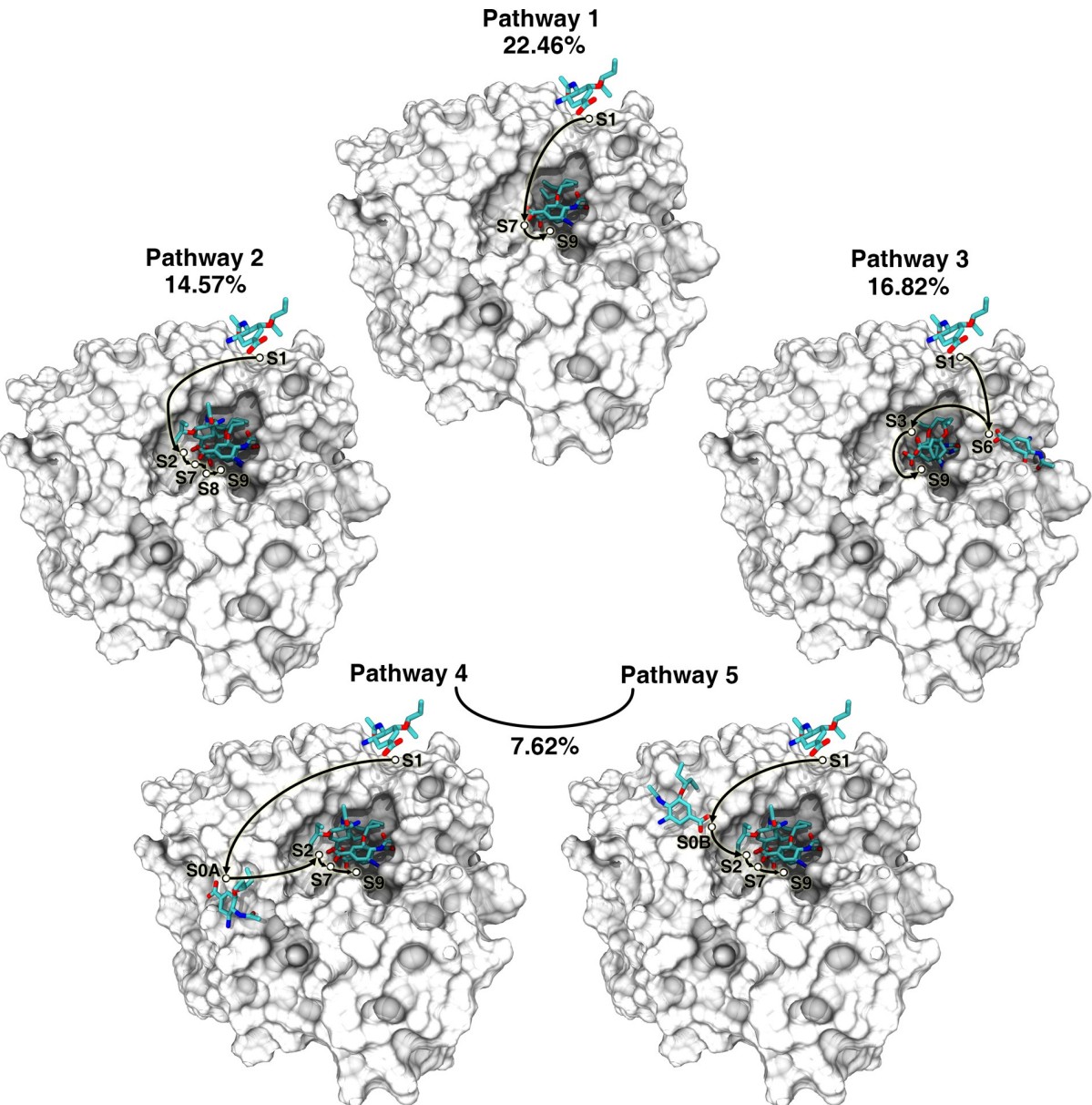

**Fig 5. Transition modes for the metastable states in the main binding pathways.** The circles refer to oseltamivir in the metastable states, and the arrows indicate transitions between the states. S1 and S9 are the start and end states of the binding process, respectively.

structure, but in our simulation, this direct H-bond was replaced by a water-mediated H-bond between them, which had not been reported previously.

To determine whether the direct H-bond in the 2QWK structure was only an exception, we systematically analyzed oseltamivir-NA interactions in all the reported cocrystal structures in RCSB (Tabel B in Section 3 in S1 Appendix) and reviewed the literature in PubMed that analyzed oseltamivir-NA interactions [14,16,22,23,41]. However, we found that there were no report on the water-mediated H-bonds between the carboxylate group of oseltamivir and the side chain of R118. However, Rudrawar *et al.* reported a sialic acid derivative (compound 5) that can bind to the NA 150 cavity, whose carboxylate group formed a water-mediated H-bond with R118 [17].

### Characterization of binding pathway

To investigate the transition of metastable states in the binding process, we used transition pathway theory (TPT) [42] to calculate the net flux and dominant reaction pathways between the unbound state (S1) and the bound state (S9) (Tables C and D in Sections 4 and 5 in S1 Appendix). The results showed there were more than 20 binding pathways (S1 Appendix, Section 5), indicating that the binding processes of oseltamivir to the active center of NA were extremely complex. In the following text, we focused on five high-flux, *i.e.* representative pathways (Fig 5). The five pathways (S1–S5 Movies) were pathway 1: S1→S7→S9, pathway 2: S1→S2→S7→S8→S9, pathway 3: S1→S6→S3→S9, pathway 4: S1→S0A→S2→S7→S9, and pathway 5: S1→S0B→S2→S7→S9.

Pathway 1 and pathway 2 had the largest (22.46%) and fourth largest (14.57%) flux proportion. In the above two pathways, oseltamivir directly entered the 1˚ site from the solvent and first formed H-bonds with R292 and R371, suggesting that R292 and R371 played an important role in capturing oseltamivir. It was reported that R292K mutation of NA conferred resistance to oseltamivir by disrupting the hydrophobic pocket that accommodated the pentyl group of oseltamivir [32], we speculated that the R292K mutation can hamper oseltamivir's entry into the 1˚ site of NA. Pathway 2 was similar to the first and only binding pathway previously reported by Zeller *et al.* [23]. Le *et al.* used steered MD (SMD) to study the dissociation process of oseltamivir and N1 NA [22], and found two dissociation pathways similar to the inverse processes of pathway 2 and pathway 5 (see below) in our study, respectively.

Pathway 3 had the second largest flux proportion (16.82%) between all pathways. Unlike pathways 1 and 2, in pathway 3, oseltamivir did not enter the 1˚ site first, but stayed temporarily outside the 1˚ site in the region around the 150 loop (S6), and was subsequently introduced into the 1˚ site by R152 on the 150 loop. Previous studies had shown that the highly flexible150 loop formed a cavity that can be a target for drug design [13–17], while our study showed that the 150 loop played an important role in introducing the substrate into the 1˚ site.

The sum of flux proportion of pathway 4 and pathway 5 was 7.62%. In pathway 5, oseltamivir entered the 1˚ site through the electrostatic funnel [22], which is the reverse process of the dissociation process of oseltamivir reported by Le *et al* [22].

Durrant *et al.* speculated that oseltamivir could be transferred from the 2˚ site to the 1˚ site along the NA surface by analyzing the distribution of chloride ions on the NA surface [13]. However, this speculation has not been directly verified until this study. In pathway 4, oseltamivir first came into contact with the 2˚ site and stayed temporarily, and then entered the 1˚ site following the introduced by R118, R371, W403, and K432. However, we speculated that the probability of this pathway was very low, because only in 2 trajectories oseltamivir moved directly from the 2˚ site to the 1˚ site and reached S9, though in 35 of the 112 positive trajectories, oseltamivir was found to be transiently trapped in the 2˚ site. In the rest 33 trajectories,

oseltamivir returned to the solvent rather than continuing to move to the 1˚ site. Although the direct transfer of oseltamivir from the 2˚ site to the 1˚ site is a rare event, we speculate that the detention of oseltamivir in the 2˚ site may increase the residence time of oseltamivir near the 1˚ site and thus increase the probability of oseltamivir entering the 1˚ site.

### Free energy and kinetics of binding process

The free-energy landscape of the binding process (Fig 6) and the mean first passage time (MFPT) for the MSM model were calculated (Table E in Section 6 in S1 Appendix) [43,44]. The free-energy landscape was generated utilizing the meta-eABF method, which reflected the free-energy change of oseltamivir on the NA surface. We projected the coordinates of metastable states in the MSM onto the free energy-landscape to obtain their free energy values (Table 1). We also used the Multidimensional Least Energy pathway (MULE) [45] to obtain the least free-energy pathways from the free-energy landscape, which corresponded to the movement of oseltamivir on the NA surface in the five pathways described above (Fig 6C). We named them pathway 1': S7→S9, pathway 2': S2→S7→S8→S9, pathway 3': S6→S3→S9, pathway 4': S0A→S2→S7→S9, pathway 5': S0B→S2→S7→S9, respectively. Our free-energy profile showed that the energetic barrier between S0A and S2 is 4.3 kcal/mol (Fig 6D), which was higher than the energetic barrier between any other two metastable state and suggested that oseltamivir was not easy to be transferred from the 2˚ site to the 1˚ site.

It is worth noting that S4 has not been reported previously and is involved in 10 pathways (Table D in Section 5 in S1 Appendix), which indicated that S4 is a key node for transition of oseltamivir to the active center of NA. Though the stability of S4 is far less stable than that of S9, our MSM results show that S4 usually transit to S2 and S7. Especially, the MFPT of S4 to S2 is 2246 ns (please see Tables C-E in S1 Appendix), which indicates that the transition speed of S4 to the S2 is slow. Therefore, we speculate that it is possible to design new drugs based on S4.

### Simulations on N9 NA tetramer

Since NA exists as a tetramer under physiological conditions, we performed 50 extra free-binding MD simulations with NA tetramer to verify whether the metastable states and binding pathways observed in the monomer simulations can be repeated. In such simulation system (600 ns each), each NA monomer "possessed" an oseltamivir molecule that was position-restricted as in the simulation with NA monomer. Twelve of the 50 simulations produced the final binding events (two oseltamivir molecules bound to the corresponding NA in two trajectories), resulting in a total of 14 positive trajectories.

By manually analyzing the positive trajectories, we found that all the metastable states and pathways 1, 2, 3, 5 occurred in the tetramer simulations. However, pathway 4 didn't occur, possibly due to its low probability of occurrence and the limited number of positive trajectories. Consistent with previous findings, we also observed slighter fluctuations in the 110 helix and 150 loop of the NA in the tetramer simulations than that in the simulations on monomer simulations [14, 46]. Furthermore, just like the simulations with NA monomer, only a water-mediated H-bond appeared between the carboxylate of oseltamivir in the bound state (S9) and the side chain of R118 in the NA tetramer simulations.

Our study showed that though under physiological conditions NA exists in the form of tetramers, NA monomer simulation can yield satisfactory results.

### Analysis of the interaction between R118 and oseltamivir

Previous study showed that in the R292K mutation the direct H-bond between oseltamivir and R292 was replaced by a water-mediated H-bond formed between oseltamivir and K292,

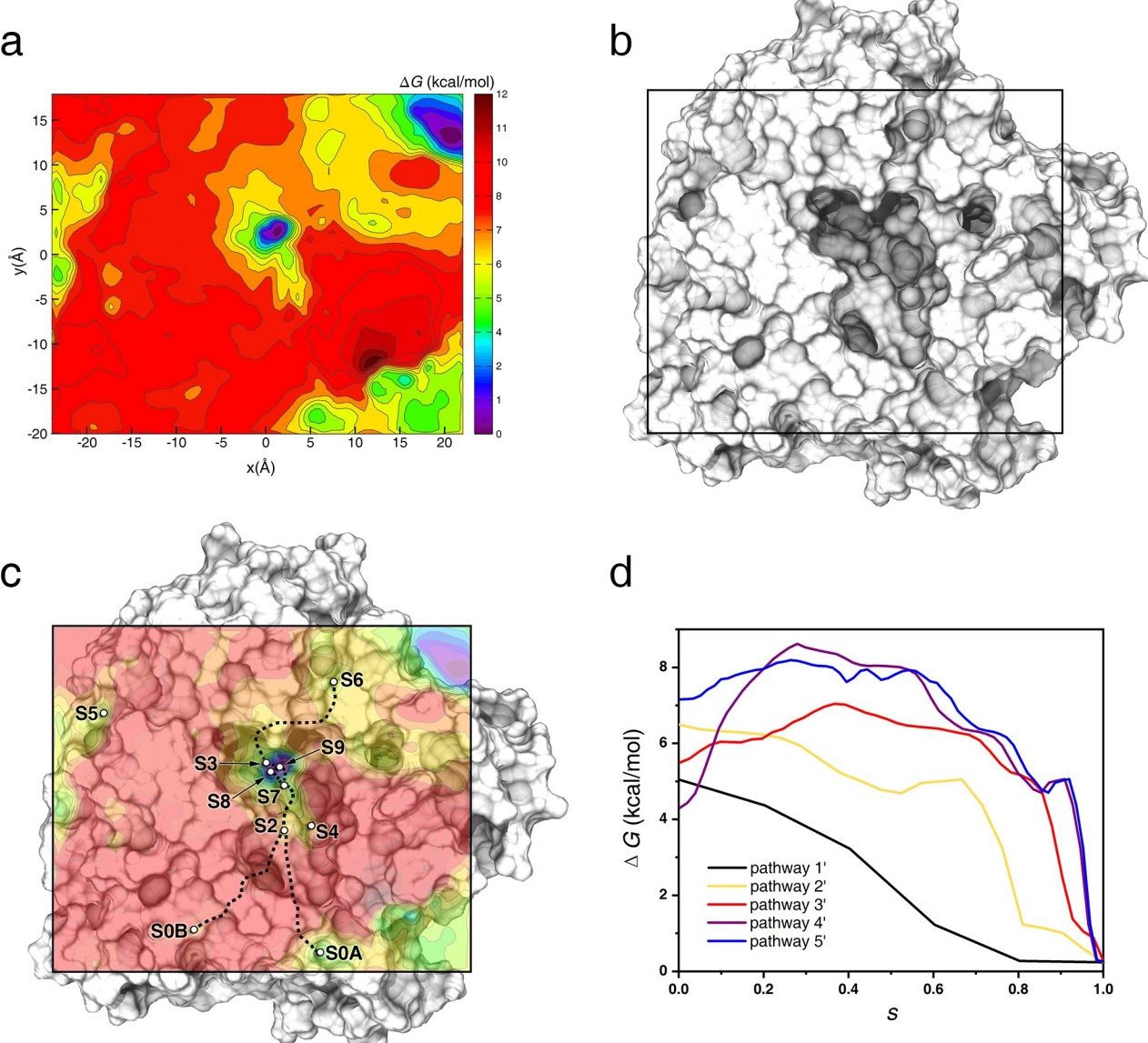

**Fig 6. Free-energy landscape and the binding pathways of oseltamivir and NA.** (a) Free-energy landscape of oseltamivir along the NA surface. Collective variables x and y represent the projection of the distance vector between the centroids of oseltamivir and NA to the x-axis and y-axis of the system, respectively. The occurrence of the region with very low free energy in the upper right corner of the free-energy landscape will be explained in S1 Appendix, Section 2. (b) Top view of the sampling space of NA in the meta-eABF simulation. (c) Fusion image of NA conformation and free-energy landscape superimposed on Fig 6A and 6B. The metastable states are represented by circles and the pathways connecting the metastable states are shown as dashed lines. (d) Free-energy profiles as a function of the position (s) along the pathways. Pathway 1': S7→S9, pathway 2': S2→S7→S8→S9, pathway 3': S6→S3→S9, pathway 4': S0A→S2→S7→S9, pathway 5': S0B→S2→S7→S9. S = 0.0 in pathways 1, 2, 3, 4 and 5 represents S7, S2, S6, S0A and S0B, respectively, and S = 1.0 in pathways 1, 2, 3, 4 and 5 represents S9.

which impaired the formation of a hydrophobic pocket (E276-R224 allosteric) accommodating the pentyl group of oseltamivir, and weakened the interaction between oseltamivir and NA [32]. In the oseltamivir-NA co-crystal structure (PDB ID: 2QWK), the carboxylate of oseltamivir interacted with the side chain of R118 by a direct H-bond, which was replaced by a water-mediated H-bond in the S9 metastable state of our MD simulations (Fig 7) performed at 37˚C. It is known that protein crystal structure is often prepared at a lower temperature than 37˚C, and protein structure gets looser at a higher temperature. Therefore the distance between

**Table 1. The free energy of metastable states.**

| Metastable state | Free energy (kcal/mol) | Metastable state | Free energy (kcal/mol) |
| --- | --- | --- | --- |
| S0A | 4.3 | S3 | 2.4 |
| S0B | 7.1 | S4 | 6.9 |
| S5 | 5.2 | S7 | 5.0 |
| S6 | 5.5 | S8 | 1.0 |
| S2 | 6.5 | S9 | 0.2 |

carboxylate of oseltamivir and the side chain of R118 increased in our MD simulations, which led to the replacement of the direct H-bond in crystal structure by the water-mediated H-bond in MD. The stability of the water-mediated H-bond between R118 and oseltamivir can be found in S5 Fig. Interestingly, in the simulation started from binding-state zanamivir-NA or peramivir-NA, the same water-mediated H-bond between R118 and zanamivir or peramivir was also observed to replace the direct H-bond in crystal structures (S6 Fig).

In order to quantitatively analyze the impact of the water-mediated H-bond on the binding strength, we used the MMPBSA approach to calculate the binding free energy of oseltamivir-NA complex in the crystal structure (PDB ID: 2QWK) and that of the final binding state in our simulations. The results of MMPBSA showed that the binding free energy of the oseltamivir-NA complex in the crystal structure (-21.7 kcal/mol) is higher than that in the binding state of our simulation (-17.0kcal/mol), and the latter is closer to the experimental value (-15.2 to -12.0 kcal/mol) [14]. The per-residue energy decomposition analysis showed that the energy contribution of R118 for oseltamivir in the crystal was -5.5 kcal/mol, however, in our simulation it was 1.0 kcal/mol, which was a huge difference (Fig 7E). The energy contributions of other residues at the 1˚ site that form H-bonds with oseltamivir to the complex were basically the same trend in the crystal structure and simulation (except for R152). Our results suggested that under physical temperature (37˚C) the direct H-bond between oseltamivir and R118 of NA in crystal structure may be changed into a water-mediated one, which may undermine the affinity between oseltamivir and NA at the 1˚ site and the therapeutic effect of oseltamivir.

## Oseltamivir-NA binding mechanisms suggest new schemes for drug design

According to our findings, the interaction between R118 and oseltamivir is weakened because the direct H-bond is replaced by the water-mediated one. Therefore, modifying oseltamivir to enhance its interaction with R118 may increase the drug affinity and therapeutic effect and counteract the impaired interaction caused by N294S, H274Y and E119V mutations in NA. Carbon-3 (C-3) of Neu5Ac2en is oriented towards the 150-cavity in a more open form of the NA, based on which Rudrawar *et al*. modified C-3 to enable it to bind to 150-cavity [17]. In our simulations, we observed that C-3 (refers to the atom name in the crystal structure) of oseltamivir is oriented towards R118, and there was a large gap between C-3 and R118. Therefore, we believe it is highly feasible to modify the C-3 of oseltamivir to enhance its interaction with the R118 of NA. Based on our result, we believe that the interaction between other NA inhibitors, like zanamivir and peramivir, with R118 is also probably to be the water-mediated H-bond rather than the direct H-bond in the crystal structure, which needs to be further investigated. We anticipate that the discovery of the water-mediated H-bond between NA inhibitors and R118 will potentially lead to the creation of the next generation of NA inhibitors.

In addition, the S4 that occurred during the binding process is also of interest. In S4, the conformation of oseltamivir is quite different from that in the bound state, but the oseltamivir shielded the catalytic residues, D151, E277 and Tyr406 [47,48]. Therefore, new NA inhibitors can be designed based on this newly discovered conformation.

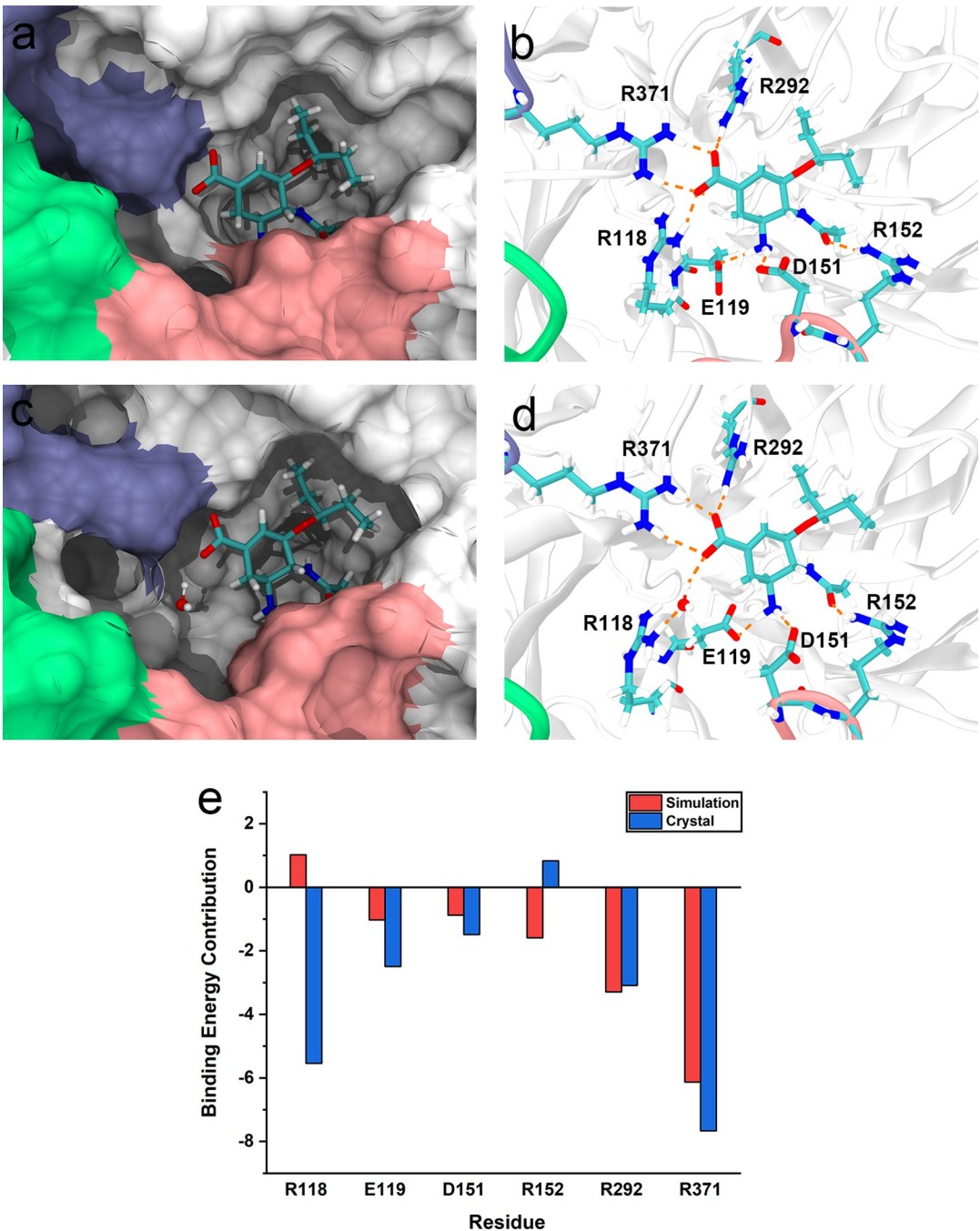

**Fig 7. Comparison of the oseltamivir-NA interactions in the crystal structure and simulation.** The H-bond is represented as an orange dotted line. (a and b) Oseltamivir-NA interaction in the crystal structure (PDB ID: 2QWK) in which oseltamivir forms a direct H-bond with R118. (c and d) Oseltamivir-NA interactions in the simulation (S9) in which oseltamivir forms a water-mediated H-bond with R118. (e) Per-residue decomposition energies for selected amino acids in the crystal structure and simulation.

## Materials and methods

### System setup for MD simulations

The crystal structure of the N9 monomer was obtained from the PDB database (PDB ID: 2QWK). All nonprotein molecules (except the $Ca^{2+}$ that coordinated with the amino acids) in

the initial crystal structure were removed. Considering that $Ca^{2+}$ is located in the vicinity of the active center and may affect the thermal stability and activity of NA [49–53], we used the MCPB.py [54] module in AMBER17 [55] to construct the bonded-model parameters for $Ca^{2+}$ (Table A in Section 1 in S1 Appendix). The protonation state of histidine and other titratable amino acids at pH 6.5 was determined by the online web service PDB2PQR [56] using the PROPKA [57] method, and the protonation state was manually confirmed.

Quantum chemical calculations were applied to prepare the parameters for oseltamivir, zanamivir and peramivir, including geometry optimization with Gaussian09 [58] at the Hartree-Fock/6-31G* level; determination of atomic charges using the RESP method [59]; and generation of the bond, angle, and dihedral parameters using the GAFF force field [60] within Amber. The ff14SB force field was applied for the protein [61].

The system was constructed by means of the LEAP module in AMBER17. There were 9 disulfide bonds in N9 NA, and CYS was annotated as CYX (the disulfide bonds of N9 NA were manually bound) to comply with the naming rules in AMBER. In the free-binding MD simulations for oseltamivir and NA, the oseltamivir molecule in the initial system was placed approximately 10 Å from the surface of NA. Sodium chloride was added to the system to a final concentration of 150 mmol/L. We constructed a truncated octahedral TIP3P water box with a 15 Å water layer for the free-binding simulation and a cubic TIP3P water box with an 18 Å water layer for the free-energy landscape calculation. More details about the system can be found in S1 Appendix, Section 1. The binding-state zanamivir-NA (PDB ID: 6HCX) or peramivir-NA (PDB ID: 4MWV) in the crystal structure was used as the starting conformation and the system was constructed according to that of oseltamivir and NA (solvated in a water box with 15 Å water layer and 150 mmol/L of sodium chloride).

Since 1–81 residues in the N-terminus of this crystal structure were missing, the residue numbers of NA in our simulations were different from the real numbers. For convenience, we provide a comparison table (Table F in Section 1 in S1 Appendix) of the residue numbers of NA between our simulation system and the PDB 2QWK crystal structure.

## Free-binding MD simulation for oseltamivir and NA

The PMEMD module of AMBER17 was used for the free-binding simulation on a graphics processing unit (GPU) [62]. To remove bad contacts between atoms, we first minimized the energy of the system before performing MD via the steepest descent method (20000 steps) and the conjugate gradient method (30000 steps). During energy minimization, we applied a restraint potential of 500 kcal·mol$^{-1}$·Å$^{-2}$ to the heavy atoms of the protein to prevent excessive energy fluctuation in the system.

After energy minimization, the system was heated from 0 to 310 K in 10 consecutive steps with an interval of 31 K. The heating time for each step was 10 ps. During this heating, we applied a gradually reduced restraint potential to the heavy atoms of the protein and the drug molecules. The limiting potential energy was 10 kcal·mol$^{-1}$·Å$^{-2}$ at the beginning. In each heating step, the restraint potential was reduced by 1 kcal·mol$^{-1}$·Å$^{-2}$. After heating, we conducted a 100 ps equilibrium for the system in the NPT ensemble at a pressure of 1 bar, a temperature of 37˚C and a step interval of 2 fs. The bond length for hydrogen atoms was limited by the shake algorithm [63]. The truncation radius for nonbonding interactions was 10 Å, and the long-range electrostatic interaction was calculated by the PME algorithm.

To accelerate the binding process of the simulations, we used position constraints to limit the diffusion space of oseltamivir. The distance between the beta carbon atom of N346 and the C5 atom on oseltamivir was used as a collective variable. When the distance between the two

atoms was greater than 27 Å (the upper bound), a penalty potential of 1 kcal·mol$^{-1}$·Å$^{-2}$ was applied to prevent oseltamivir molecules from escaping from the limited region.

We also performed free-binding simulations with the NA tetramer to compare the binding process with those of the NA monomer. The NA tetramer was prepared by matrix transformation starting from the 2QWK structure. Four oseltamivir molecules were added to the system. The position restriction of oseltamivir and the simulation parameters were the same as those in the simulation with the NA monomer.

### Meta-eABF simulation

To obtain the free-energy landscape of the binding process of oseltamivir-NA, we used NAMD software to conduct a 16 μs meta-eABF simulation. The sampling space was limited to a 46 Å×38 Å×32 Å box, which included the 1° and 2° sites and their adjacent regions (Fig 1B and 1C). The center of the lower xy plane of this box was set as the centroid of NA (just within the 1° site). Collective variables x and y represent the projection of the distance vector between the centroids of oseltamivir and NA to the x-axis and y-axis of the system, respectively. For more information about the meta-eABF simulation, please see S1 Appendix, Section 2.

### Analysis

The Visual Molecular Dynamics (VMD) software [64] was used to create pictures of the 3D protein and oseltamivir molecule. The root mean square deviation (RMSD) for heavy atoms of oseltamivir in NA monomer simulation trajectories with respect to that in the 2QWK structure was calculated at 0.1 ns intervals by RMSD Trajectory Tool in the VMD. The binding free energies and per-residue energy decomposition were evaluated using AMBER program MMPBSA [65]. The hydrogen atoms of the crystal structure (PDB ID: 2QWK) were added using PROTOSS [66,67] when using MMPBSA.

Markov State Models (MSMs) of the binding process were constructed from 112 positive trajectories of NA monomer simulations with trajectory frames taken every 0.018 ns. To describe the binding between oseltamivir and NA, we used nearest-neighbor heavy-atom contacts between the oseltamivir residue and each other residue as the input for model construction. Trajectory frames were clustered into 200 microstates by k-means clustering, as implemented in the PyEMMA software package. Subsequently, we lumped our model into 10 metastable states using the PCCA++ and hidden Markov state models (HMMs). Based on the implied timescale plots (S2 Fig), a lag time of 2.7 ns was selected for model construction. The resulting models were validated by the Chapman-Kolmogorov (CK) test, which gives a sense of the 'correctness' and convergence of the HMMs (S3 Fig). Confidence intervals were calculated using Bayesian hidden Markov state models (BHMMS) corresponding to the HMMs as described above.

To obtain the free-energy profiles of the main binding pathways, we used the Multidimensional Least Energy pathway (MULE). We set the coordinates of the start point of the pathways as the coordinates of the projection of S2, S7, S6, S0A, and S0B on the free-energy landscape, and the coordinates of the end point were the projected coordinates of S9.

### Supporting information

**S1 Fig. Interaction diagram between oseltamivir and NA in metastable states.** Ligplot+ was used to plot the interaction between oseltamivir and NA in metastable states. Hydrogen bonds are shown as green dotted lines. Hydrophobic contacts are shown as red dotted lines. (PDF)

**S2 Fig. Implied time scale plots.** Implied time scale plots with errors as a function of lag time for the HMMs constructed based on the simulation data. Each colored line represents the timescales of different dynamical processes (motions) identified by the decomposition of the transition matrix (eigenvalues). If the model was Markovian (at the chosen lag time), then the timescales would be constant for all longer lag times that were also short enough to resolve the process. Bayesian errors are indicated as similarly colored shaded areas.
(PDF)

**S3 Fig. Chapman-Kolmogorov test for the HMMs.** The plots indicate convergence of the presented models.
(PDF)

**S4 Fig. The bonded model for Ca$^{2+}$-coordinated residues.**
(PDF)

**S5 Fig. The stability of oseltamivir-NA complex and water-mediated H-bond.** The simulations were performed with oseltamivir-NA complex of NA tetramer starting from the crystal structure (PDB ID: 2QWK). (a) The RMSD values of oseltamivir and (b) NA. (c) Time evolution of distance between the O1B atom of oseltamivir and NH1 atom of Arg118 in the 16 individual oseltamivir-NA complex. (d) Column 1–16 is the average distance between the O1B atom of oseltamivir and NH1 atom of Arg118 of individual oseltamivir-NA complex (4.64 Å ~ 6.84 Å), column 17 is the average distance between the O1B atom of oseltamivir and NH1 atom of Arg118 of all oseltamivir-NA complex (5.96Å).
(TIF)

**S6 Fig. The water-mediated H-bond between R118 and (a) zanamivir or (b) peramivir.**
(TIF)

**S1 Appendix. Seven sections providing additional information.** Section 1: Detailed parameters for molecular dynamics (MD) simulations. Section 2: Detailed parameters of the meta-eABF simulation. Section 3: Analysis of the H-bond between the carboxylate group of oseltamivir and the guanidine group in the crystal structure containing oseltamivir and influenza NA. Section 4: Transition path theory fluxes from the unbound state (S1) to the bound state (S9). Section 5: Pathway analysis based on metastable states. Section 6: Mean first passage time between metastable states. Section 7: Comparison of residue numbers of NA between our simulation system and the PDB 2QWK crystal structure. Table A. Partial Charge and Atom Type of the Metal Center. Table B. Basic information about the oseltamivir-NA crystal structure. Table C. Net flux ($\times 10^{-6}$/ns) from the unbound state (S1) to the bound state (S9). Table D. Dominant reaction pathways between the unbound state (S1) and the bound state (S9). Table E. Mean first passage time between metastable states. Table F. Comparison of residue numbers of NA between our simulation system and the crystal structure PDB 2QWK.
(DOCX)

**S1 Movie. Oseltamivir-neuraminidase pathway 1(S1→S7→S9).** The detours and recrossings of the transition of metastable states were removed in movies.
(MP4)

**S2 Movie. Oseltamivir-neuraminidase pathway 2(S1→S2→S7→S8→S9).**
(MP4)

**S3 Movie. Oseltamivir-neuraminidase pathway 3(S1→S6→S3→S9).**
(MP4)

**S4 Movie. Oseltamivir-neuraminidase pathway 4(S1→S0A→S2→S7→S9).**
(MP4)

**S5 Movie. Oseltamivir-neuraminidase pathway 5(S1→S0B→S2→S7→S9).**
(MP4)

**S1 Data. MD input files.** Input files for oseltamivir-NA monomer simulation, oseltamivir-NA tetramer free binding simulation, oseltamivir-NA tetramer binding state complex simulation, zanamivir-NA tetramer binding state complex simulation, peramivir-NA tetramer binding state complex simulation and meta-eABF simulation.
(ZIP)

**S2 Data. PDB files of metastable states.** PDB files of typical conformation of the metastable states identified from our MSM (with water molecules and counter ions removed). A PDB file (containing water molecules and counter ions) of the metastable state S9 (corresponding to the NA subunit of residues 1~389) appearing in the oseltamivir-NA tetramer simulation.
(ZIP)

## Author Contributions

**Conceptualization:** Jiaye Tao, Xinguang Liu, Zuguo Zhao.

**Data curation:** Jiaye Tao, Jiajing Ouyang.

**Formal analysis:** Jiaye Tao, Jiajing Ouyang.

**Funding acquisition:** Wenjian Wang, Xuemeng Li, Zhiwei He.

**Investigation:** Jiaye Tao, Jiajing Ouyang.

**Methodology:** Jiaye Tao, Zuguo Zhao.

**Project administration:** Heping Wang, Xinyun Liang, Zuguo Zhao.

**Resources:** Guoming Li, Shengjun Feng, Zuguo Zhao.

**Software:** Jiaye Tao, Jiajing Ouyang, Zuguo Zhao.

**Supervision:** Na Mi, Zuguo Zhao.

**Validation:** Na Mi, Wei Zhang, Min Chen, Wentao Guo, Jun Liu.

**Visualization:** Jiaye Tao, Qiujia Wen, Jiajing Ouyang, Hanning Zhao, Xin Wang.

**Writing – original draft:** Jiaye Tao, Jiajing Ouyang, Zuguo Zhao.

**Writing – review & editing:** Jiaye Tao, Na Mi, Zuguo Zhao.

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
