## [Decision Letter · Decision Letter 0]

6 Jun 2022

Dear Dr. Zhao,

Thank you very much for submitting your manuscript "Binding mechanism of oseltamivir and influenza neuraminidase suggests perspectives for the design of new anti-influenza drugs" for consideration at PLOS Computational Biology.

As with all papers reviewed by the journal, your manuscript was reviewed by members of the editorial board and by several independent reviewers. In light of the reviews (below this email), we would like to invite the resubmission of a significantly-revised version that takes into account the reviewers' comments.

We cannot make any decision about publication until we have seen the revised manuscript and your response to the reviewers' comments. Your revised manuscript is also likely to be sent to reviewers for further evaluation.

Sincerely,

Alexander MacKerell

Associate Editor

PLOS Computational Biology

Nir Ben-Tal

Deputy Editor

PLOS Computational Biology

Reviewer's Responses to Questions

**Comments to the Authors:**

Reviewer #1: It's an interesting and a comprehensive piece of work.

Following are my comments:

Although 10 meta-stable states were identified using 'constrained' free(?) MD using ligand at a distance from the binding, it would have been appropriate to show the stability of protein-ligand complex in MD starting from the crystal structure and whether a water-mediate H-bond between ligand and Arg118 is observed in MD using the bound(crystal) state. This could have served as an additional validation.

It would be difficult to isolate the effect of the force field (and simulation parameters) for the Metastable states S3, S4, S8 and S9 where Oseltamivir adopts alternate bound configurations compared to the one observed in the crystal structure. It has been challenging to obtain a stable protein-ligand complex between Neuraminidase (and similar viral proteins with 3 Arg in the binding site) and ligand in long MD simulations as observed from published (10.26434/chemrxiv-2021-56ds7), especially for monomers (compared to tetramer for neuraminidase).

It would be good to see the stability of the water-mediated complex between NA (Arg118) and Oseltamivir in longer MD simulations.

Additional findings suggest that metastable state S4, the ligand adopts an alternate bound conformation compared to the crystal structure state. Thermodynamic stability/preference between these two states (not given in the paper) would be useful to choose one or both states for drug design.

Other Meta-stable states and binding pathways have less bearing on the design of better inhibitors.

Reviewer #2: In this study, the authors perform a large number of molecular dynamics (MD) simulations of neuraminidase (NA) monomers and tetramers with the oseltamivir inhibitor for the purpose of performing a Markov state model (MSM) analysis of various binding and metastable states. The authors also perform a number of free energy and kinetics calculations, and suggest some chemical modifications to existing NA inhibitors that, the authors allege, may enable the production of new anti-influenza drugs. The authors identify nine states, which define the unbound state, the active site and secondary site bound states, and a number of other metastable states, all of which interact with certain key residues and loops that have (and on occasion have not) been previously identified in the literature. Significantly, the authors find that a relatively high free energy barrier exists between the active site and the secondary site of NA for oseltamivir, opposing the hypothesis that the secondary site acts as a sort of bridge from the solvent to the active site. In addition, the authors focus on the large gap, likely a water-mediated hydrogen bond, between existing inhibitors, including oseltamivir, and residue R118. The authors believe that this gap represents an ideal drug design opportunity.

This reviewer believes that the authors have performed a significant and meaningful study of the binding mechanisms between NA and oseltamivir, and that this manuscript contains valuable and interesting information for the broader scientific community. However, minor revisions are suggested before acceptance for publication, and the manuscript has a number of weaknesses, especially spelling and grammar mistakes, which undermine what would otherwise seem to be a valid study.

Minor Revisions:

The first sentence of the manuscript on line 65 of page 4 “Among influenza viruses, influenza A is the most harmful” is unsupported by references or further information. It is also unclear how this statement fits with the rest of the study (is the study performed on an influenza A neuraminidase?).

On line 82 of page 4, “crystal structures of oseltamivir-NA have been resolved” should be clearer by saying “ crystal structures of oseltamivir-NA holo complexes have been resolved”.

The authors use the “position restraint method” to keep the ligand within 27 Angstroms of the active site of NA. What is this method? Is this a harmonic function or step function in the potential energy? If this is a force, what is its mathematical form? If it is not a force, then how exactly does this work?

The authors may consider using “MD” instead of “molecular dynamics” throughout the manuscript for brevity.

In the Fig. 1 caption, the authors say, “the CB of N346”. They should explicitly say “beta carbon” instead of CB, to be clearer.

On line 138 of page 7, the authors say: “[the different binding times observed in the trajectories] indicate that the oseltamivir-NA binding process is complex and there may be multiple binding pathways”. This is not a valid statement. It is perfectly possible to obtain an entire distribution of observed trajectory times for a molecular process due to Brownian motion, even if there is only one pathway. The evidence for multiple pathways in this study, rather, comes from the MSM results.

Page 8 line 154 misspelled “below”. (Lots of spelling and grammar errors, the authors should check their spelling and grammar carefully.)

Page 10 line 186, should the authors say “dotted square” instead of “dotted line”?

Page 16 line 296 has an incomplete sentence.

If possible, the authors should add error margins to Table 1.

Page 20, line 352, the word “recurred” should probably be “repeated”.

Page 21, line 362, misspelled “simulations”

Page 21, lines 366 to 368 has a different font than the rest of the manuscript.

Page 21, line 370 misspelled “destroyed”.

The entire “Analysis of the interaction between R118 and oseltamivir is full of more spelling and grammar errors - too many to list here.

Please define the meaning of “physical temperature” one page 21, line 377. Is there something significant about 37 degrees celsius?

Page 24, lines 422 to 425 have two redundant sentences that are probably saying the same thing.

Page 25, line 441, why are the authors choosing to use MCPB.py for a calcium ion? Unless the metal is covalently bound to the protein, MCPB.py is unnecessary, and the authors can just use the ordinary calcium ion parameters. If the calcium is covalently bound, then the calcium cannot be an ion. The authors should explain why they used MCPB.py, whether the calcium is an ion, and why this choice will not damage the validity of the MD simulations.

The authors should explain better why NAMD was used for the free energy landscape, and not AMBER, as with the other simulations. It is unusual to use two separate MD simulation programs for the same study.

**Have the authors made all data and (if applicable) computational code underlying the findings in their manuscript fully available?**

Reviewer #1: Yes

Reviewer #2: None

PLOS authors have the option to publish the peer review history of their article (what does this mean?). If published, this will include your full peer review and any attached files.

Reviewer #1: No

Reviewer #2: **Yes: **Rommie Amaro
---

## [Editor Report · Decision Letter 1]

30 Jun 2022

Dear Dr. Zhao,

We are pleased to inform you that your manuscript 'Binding mechanism of oseltamivir and influenza neuraminidase suggests perspectives for the design of new anti-influenza drugs' has been provisionally accepted for publication in PLOS Computational Biology.

Best regards,

Alexander MacKerell

Associate Editor

PLOS Computational Biology

Nir Ben-Tal

Deputy Editor

PLOS Computational Biology

---

## [Editor Report · Acceptance letter]

24 Jul 2022

PCOMPBIOL-D-22-00626R1 

Binding mechanism of oseltamivir and influenza neuraminidase suggests perspectives for the design of new anti-influenza drugs

Dear Dr Zhao,

I am pleased to inform you that your manuscript has been formally accepted for publication in PLOS Computational Biology. Your manuscript is now with our production department and you will be notified of the publication date in due course.

With kind regards,

Olena Szabo
